# GWAS on Imputed Whole-Genome Sequence Variants Reveal Genes Associated with Resistance to *Piscirickettsia salmonis* in Rainbow Trout (*Oncorhynchus mykiss*)

**DOI:** 10.3390/genes14010114

**Published:** 2022-12-30

**Authors:** Charles Sánchez-Roncancio, Baltasar García, Jousepth Gallardo-Hidalgo, José M. Yáñez

**Affiliations:** 1Doctorado en Acuicultura, Programa Cooperativo: Universidad de Chile. Universidad Católica del Norte. Pontificia Universidad Católica de Valparaíso, Chile; 2Center for Research and Innovation in Aquaculture (CRIA), Universidad de Chile, Santiago 8820808, Chile; 3Facultad de Ciencias Veterinarias y Pecuarias, Universidad de Chile, La Pintana, Santiago 8820808, Chile; 4Núcleo Milenio de Salmonidos Invasores Australes (INVASAL), Concepcion 4030000, Chile

**Keywords:** *Piscirickettsia salmonis*, GWAS, whole genome sequencing, rainbow trout

## Abstract

Genome-wide association studies (GWAS) allow the identification of associations between genetic variants and important phenotypes in domestic animals, including disease-resistance traits. Whole Genome Sequencing (WGS) data can help increase the resolution and statistical power of association mapping. Here, we conduced GWAS to asses he facultative intracellular bacterium *Piscirickettsia salmonis*, which affects farmed rainbow trout, *Oncorhynchus mykiss*, in Chile using imputed genotypes at the sequence level and searched for candidate genes located in genomic regions associated with the trait. A total of 2130 rainbow trout were intraperitoneally challenged with *P. salmonis* under controlled conditions and genotyped using a 57K single nucleotide polymorphism (SNP) panel. Genotype imputation was performed in all the genotyped animals using WGS data from 102 individuals. A total of 488,979 imputed WGS variants were available in the 2130 individuals after quality control. GWAS revealed genome-wide significant quantitative trait loci (QTL) in Omy02, Omy03, Omy25, Omy26 and Omy27 for time to death and in Omy26 for binary survival. Twenty-four (24) candidate genes associated with *P. salmonis* resistance were identified, which were mainly related to phagocytosis, innate immune response, inflammation, oxidative response, lipid metabolism and apoptotic process. Our results provide further knowledge on the genetic variants and genes associated with resistance to intracellular bacterial infection in rainbow trout.

## 1. Introduction

In 2010, the world production of rainbow trout was 752,400 tons, and in 2020, it increased to 960,600 tons; a very rapid growth of almost 27% in ten years [1]. In 2020, Chile harvested around 85,900 tons of rainbow trout [2]. The production of this species is highly affected by infectious diseases, such as piscirickettsiosis or Salmonid Rickettsial Syndrome (SRS) caused by *P. salmonis*, producing the greatest economic losses. In the first semester of 2022, 8.5% of the total mortality was attributed to SRS [3]. Prophylactic and control strategies against *P. salmonis* infection have been based mainly on vaccines and antibiotic administration during disease outbreaks. However, both approaches show low effectiveness in controlling SRS in field conditions [4,5]. During the first semester of 2022, 94.8% of the antibiotics administered to salmonids in Chile were for the treatment of *P. salmonis* infection [6]. The use of antibiotics might result in resistant bacterial strains, not only reducing the efficacy of treatments and affecting the success and sustainability of aquaculture but also representing an important threat to public health [7,8]. The limited efficacy of vaccines on reducing mortalities in sea-cages might be due to the variation in the host immune response between families and the complexity of the *P. salmonis*–host interaction [9,10,11,12,13,14,15]. An alternative strategy to reduce the detrimental effects of this disease in the long term, is the genetic selection of fish more resistant to *P. salmonis* infection [16,17,18]. In fact, the most important salmon and trout breeding companies in Chile currently offer SRS-resistant eggs within their products, which are generated from families which have shown superior survival in either experimental or field conditions.

With the advances in the genomics of aquatic organisms in recent years [18], it is now possible to include information from dense molecular markers (e.g., single nucleotide polymorphisms (SNPs), in aquaculture breeding programs [19]. Genomic information has two main applications for breeding programs: genomic selection and genome-wide association studies (GWAS). The genomic selection allows the estimation of genomic breeding values of selection candidates with greater accuracy than pedigree-based methods [20], which accelerates the genetic progress for the trait of interest. For example, the effect of genomic selection in increasing the accuracy on the estimation of the genetic merit for resistance to *P. salmonis* infection has been demonstrated in Atlantic Salmon [21], coho Salmon [22] and rainbow trout [18,23].

The GWAS aims to map quantitative trait loci (QTL) to understand the genetic architecture and find DNA variants associated with the trait of interest [24]. This approach is based on the evaluation of the association between genotypes and phenotypes, using different statistical methods [25]. Several factors may influence the accuracy of GWAS such as the heritability of the trait, genetic architecture, statistical method for the association, number of samples (n), and number of SNPs [26,27,28,29,30]. The inclusion of whole-genome sequence (WGS) genotypes might increase the accuracy of genomic prediction and QTL mapping compared to lower-density SNP arrays, especially if causal genetic variants are included in the analysis [31,32,33]. Despite the drastic reduction in WGS costs in recent years, it is still cheaper to genotype animals using SNP chips than re-sequencing individuals at the whole-genome level [34]. An interesting approach to reduce the cost of obtaining WGS individual data is the imputation of genotypes, which consists of predicting unknown genotypes in animals genotyped with a panel of SNPs of a lower density using animals genotyped with a panel of denser SNPs or whole-genome sequenced as reference [35]. The imputation of SNP genotypes to WGS has been performed in pigs [36,37], poultry [38,39], cattle [34,40,41], rabbits [42], rainbow trout [43] and Nile tilapia [44,45] as a strategy to increase the number of SNPs to boost statistical power and accuracy of association between markers and QTLs [33].

Mapping of QTLs through GWAS has been used to identify genomic regions related to disease resistance in different fish species [19]. GWAS for resistance against different diseases, including *Flavobacterium psychrophilum* [46,47,48], *Aeromonas salmonicida subsp. salmonicida* [49], *Vibrio anguillarum* [50], *Ichthyophthirius multifiliis* [51], IPN virus [52], *Flavobacterium columnare* [53] and *Yersinia ruckeri* [54], have been performed in rainbow trout using a medium density (57K) SNP chip. Potential QTLs associated with *P. salmonis* resistance have been found in rainbow trout and other salmonid species [21,22,23,55,56]. For instance, resistance to *P. salmonis* infection was determined to be a polygenic trait in rainbow trout, Atlantic salmon and coho salmon, with few QTL explaining a moderate proportion of the genetic variance for the trait [22,23,55]. All these studies were performed using medium-density SNP panels, including tens of thousands of markers, making it difficult to map genomic regions with high accuracy. This study aimed to perform GWAS for resistance against SRS using genotype data imputed to high-density SNP information (i.e., WGS) for increasing statistical power and resolution in the detection of genomic regions and putative genes controlling the trait.

## 2. Materials and Methods

### 2.1. Origin of Animals and Challenge to P. salmonis

The animals used in this study belonged to the 2011 year-class from the rainbow trout breeding nucleus owned by EFFIGEN S.A. (Puerto Montt, Chile). This breeding program has applied selection for different traits of interest (growth, carcass quality and appearance) since 2001. For the experimental challenge against *P. salmonis*, juveniles from 102 families, weighing approximately 7.0 ± 1.5 g were individually tagged with a PIT-tag, kept for seven months in a single tank, and then transferred to the Aquainnovo Research Station (X Region, Chile). The fish had an acclimatization period of 20 days before the challenge. Random sampling was performed to check for Flavobacterium spp., Infectious Pancreatic Necrosis virus (IPNV), infectious salmon anemia virus (ISAV), and *Renibacterium salmoninarum* using qRT-PCR. Then, 2130 juveniles (an average of 20 individuals per family, ranging from 12 to 27 fish) were inoculated intraperitoneally (IP) with 0.2 mL of the *P. salmonis* strain LF-89 at a lethal dose of 50, as described by [23]. After inoculation, the fish were distributed equally in three tanks keeping five to nine fish per family in each tank. The challenge lasted 32 days, in which daily mortality and final weight were recorded at death or at the end of the experiment if the fish survived. Survivor animals were euthanized, and their body weight were also recorded. Tissue samples were taken from the caudal fin to isolate the genomic DNA of all challenged fish and stored in 95% ethanol at −80 °C for subsequent genotyping.

Resistance against SRS was evaluated using the time of death (TD) with values ranging from day 10 to day 32 and binary survival (BS), where 1 indicates that the fish died and 0 that it survived challenge.

### 2.2. Genotyping of Challenged Animals

Genomic DNA was extracted from the 2130 sampled fins using a commercial kit (DNeasy Blood & Tissue Kit, Qiagen, China), following the manufacturer’s instructions. Individual genotypes were obtained using a commercial 57K SNP chip (*Affymetrix Axiom*) developed by the National Center for Cool and Cold Water Aquaculture of the USDA Agricultural Research Service [57]. The quality control (QC) of the genotypes was applied through the Plink v1.90 software [58], with a call rate for SNP and sample higher than 0.90, minor allele frequency (MAF) threshold of 0.01 and Hardy–Weinberg equilibrium (HWE) of *p*-value > $10^{-8}$.

### 2.3. Whole-Genome Sequence Data

Detailed information on the sequencing procedures is available in [43]. Briefly, genomic DNA was extracted from the fin-clips of 102 individuals using the DNeasy Blood & Tissue kit (Qiagen) according to the manufacturer’s instructions. The samples were sent to the Beijing Genomics Institute (BGI, China) for whole genome sequencing using DNBseq technology. The *O. mykiss* genome (GenBank Assembly Accession GCA_013265735.3 USDA_OmykA_1.1) was used to align re-sequencing data. This reference genome version is comprised of a final assembly length of ~2.33 Gbp, 1591 contigs with an N50 contig length of 9, 8 Mb, of which ~95% is anchored in 29 chromosomal sequences. The re-sequencing reads from each sample were mapped to the reference genome using the Burrows-Wheeler Aligner (BWA) analysis tool [59], resulting in a mapping rate and effective mapping depth between 97.51 and 98.16%, and 10.31× and 17.65×, respectively. A protocol implemented in the Genome Analysis Toolkit (GATK, https://www.broadinstitute.org/gatk/, accessed on 19 June 2021) was used for the SNP calling. The final VCF file consisted of 22.6 million non-redundant variants present in the 102 rainbow trout re-sequenced individuals. The quality control of genotypes was performed with Plink v1.90 [58] using the following criteria: SNP and sample call-rate higher than 0.80, MAF > 0.01 y HWE *p*-value > $10^{-8}$.

### 2.4. Imputation to Whole Genome Sequence Level

For the validation of the imputation, five cross-validation sets were formed using only the 102 parents divided into 20% validation (approximately 20 animals per group) and 80% reference (approximately 80 animals per group) sets. The validation animals had only the 57K SNPs in common with the genotyped animals (challenged animals)and the remaining SNPs were masked to perform genotype imputation. Reference animals had information at WGS level including only SNPs that passed by QC. FImpute v3.0 [60] was used to perform all genotype imputations. Accuracy was assessed using the r^2^ (Pearson’s squared correlation between observed and imputed genotypes) statistic, and SNPs with low imputation accuracy (lower than 0.8) were removed from the final set of imputable SNPs. Finally, the challenged animals genotyped with 57k SNPs were imputed using only validated SNPs with an accuracy above 0.8.

### 2.5. Genome-Wide Association Study (GWAS)

GWAS was performed for two trait definitions: time to death (TD) and binary survival (BS) using a linear mixed animal model implemented in the Genome Complex Trait Analysis (GCTA) software. This software uses a Restricted Maximum Likelihood (REML) analysis to estimate the proportion of additive genetic variation captured from all SNPs [58] through the *mlma* function. The linear mixed model applied was:Y=Xβ+Zα+e
where Y is a vector of phenotypes (TD and BS), β is the vector of fixed effects (tank as fixed effect and body weight measured at the end of the experiment as covariate), α is a vector of random additive polygenic genetic effects with distribution ∼N(0, ${G\sigma }_{a}^{2}$), where $\sigma_{a}^{2}\;$ is the additive genetic variance and G is the genomic relationship matrix (GRM) calculated using the imputed genotypes [61], e is the vector of residuals with distribution ∼N(0, ${I\sigma }_{e}^{2}$), where I is the identity matrix, and $\sigma_{e}^{2}$ is the residual variance, and X and Z are the incidence matrices for fixed and random effects, respectively. The allelic substitution effect and *p*-value for each SNP were also estimated using the GCTA.

For TD and BS, heritability ($h^{2}$) was estimated as follows:$$h^{2}=\frac{\sigma_{a}^{2}}{\sigma_{a}^{2}+\sigma_{e}^{2}\;}$$
where $\sigma_{a}^{2}$ is the estimated additive genetic variance using the GRM, and $\sigma_{e}^{2}$ is the residual variance.

### 2.6. Identification of Candidate Genes

The SNPs significantly associated with *P. salmonis* resistance were identified on the basis of the genomic significance threshold (0.05/number of SNPs) and the chromosomal significance threshold (0.05/number of SNPs), which represent α values corrected by Bonferroni. A systematic approach to search for candidate genes was performed using the Multi-marker Analysis of Genomic Annotation (MAGMA) software, which uses a multiple regression model to test the joint effect of multiple markers of a gene and then tests the association of the gene with the trait [62]. The Omyk_1.0 reference genome of *O. mykiss* (GenBank Assembly Accession GCA_002163495.1) was used in the gene search, and Bonferroni correction was also applied to obtain the significance threshold of the candidate genes.

## 3. Results

### 3.1. Summary Statistics

Similar mean values were observed for TD: 24.55 (SD = 7.60); 23.5 (SD = 7.82) and 22.73 (SD = 8.08) days for challenge tanks 1, 2, and 3, respectively. A Kaplan–Meier survival curve [63] was drawn for the best and worst families, as well as the average between families (Figure 1). Our results demonstrated significant (*p* > 0.05) phenotypic variation based on Kaplan–Meier survival analysis. For final weight (FW), values were: 177.5 (SD = 52.05); 167.3 (SD = 44.90) and 176.2 (SD = 58.62) grams for challenge tanks 1, 2 and 3, respectively (Figure 2).

### 3.2. Quality Control of Genotypes and Imputation

A 57K SNP chip was used for genotyping 2130 challenged rainbow trout. Initial quality control (QC) of genotypes was applied with Affymetrix Axiom software according to default settings. Additional QC filters were sequentially applied, discarding 17.6% SNPs by call-rate, 2.7% SNPs by MAF and 17.14% by HWE, leaving a total of 31,275 SNPs available for downstream analyses. For the genotypes from WGS data, 22,649,022 SNPs from the 102 samples were obtained after SNP calling. The call-rate filter eliminated about 83.3% of the initial SNPs. MAF and HWE filters discarded 5.2% and 1% of SNPs, respectively, with a remaining list of 2,382,000 SNPs available for downstream analysis (a total of 10.5% out of the initial number of SNPs. After imputation validation, 24.3% of the SNPs had an r² value equal or greater than 0.80 and were used as the reference data set for the definitive imputation round. A post-imputation quality control (HWE < *p*-value > $10^{-8}$ and MAF < 0.05) resulted in a total of 488,979 available imputed SNPs for the 2130 individuals reported as imputed WGS genotypes (Table 1).

### 3.3. Genome-Wide Association Studies

Genetic parameters were estimated using the 488,979 imputed genotypes. The estimated additive genetic variance was 13.59 and 0.045, and the heritability values were 0.27 (SD = 0.02) and 0.25 (SD = 0.02), for TD and BS, respectively (Table 2).

GWAS for resistance to *P. salmonis* measured as TD and BS was performed for each trait using the imputed WGS data (Figure 3). Five genomic regions were found to be genome-wide and significantly associated with TD and one region for BS. Genome-wide significant QTLs for TD were found in *Omy2*, *Omy3*, *Omy25*, *Omy26* and *Omy27*, while for BS the only significant QTL was found in *Omy26*. In addition, chromosome-wide significant QTLs for TD were found in *Omy3*, *Omy06*, *Omy10*, *Omy13*, *Omy14*, *Omy17* and *Omy28* while for BS, only suggestive QTL were found in *Omy3*, *Omy17*, *Omy26* and *Omy27*. All QTLs found here explained a low phenotypic variance for resistance against *P. salmonis*, reinforcing the fact that this trait is under polygenic control in rainbow trout (Appendix A).

### 3.4. Gene Identification

A total of 20,980 genes were available for a downstream gene search using MAGMA [62]. Table 3 shows the 24 candidate genes which were identified to be related to *P. salmonis* resistance, according to the Bonferroni-corrected significance threshold (0.05/20,980 = $2.38\times 10^{-6}$ used for MAGMA analysis. Most of the identified genes are potentially involved in important biological functions related to the host response to *P. salmonis* infection, such as phagocytosis, innate immune response, inflammation, oxidative response, protein signaling pathway G, lipid metabolism, and the apoptotic process (Table 3).

## 4. Discussion

IP challenges have been extensively used to assess genetic variation for resistance to SRS given the high level of replication between experiments. In addition, the genetic correlation between IP and cohabitation challenges has been shown to be >0.8 [64,65] indicating that resistance to *P. salmonis* experimentally evaluated by IP or cohabitation is similar traits, from the genetic perspective. The estimated SNP-heritability for TD and BS using WGS-imputed genotypes were 0.27 ± 0.02 and 0.25 ± 0.02, respectively. These estimates were lower than those presented by [23], which were calculated in the same population using a different statistical approach implemented in BLUPF90 software, with heritability values of 0.48 ± 0.04 and 0.34 ± 0.04 for TD and BS, respectively. Here, we used a different density of SNPs compared with the previous study, which by imputation error may have had an impact on the estimation of genetic parameters. However, the results obtained are similar to those obtained when pedigree and genomic information was simultaneously used to estimate the genetic variance for resistance against *P. salmonis* in rainbow trout [18,23]. In general, our results confirm that there is a significant additive-genetic component involved in phenotypic variance for *P. salmonis* resistance.

Consistent with the results presented here, previous association-mapping studies in different salmonid species, indicate that resistance to *P. salmonis* is a polygenic trait [22,23,55]. Most of the previous works were performed through GWAS using low and medium-density SNPs [23,56,66]. Here, we found numerous SNPs significantly associated with *P. salmonis* resistance (Figure 1), most likely due to the polygenic architecture of this trait in rainbow trout, together with the reasonable sample size and increased SNP density. Although imputation or genotyping errors may generate false positives of association, the stringent filters used for the QC of true and WGS-imputed genotypes aimed at decreasing the probability of false positive detection. In addition, imputation results are in agreement with previous studies which performed genotype imputation to WGS in rainbow trout and other aquaculture species [43,44,45]. It is noteworthy that the most important QTL for TD, located in *Omy27* and including several SNPs surpassing the genome-wide significance threshold, is in agreement with results from a previous study in which the same genomic region explained the highest proportion of the genetic variance for *P. salmonis* resistance [23]. The significant and suggestive QTLs identified for TD and BS in *Omy3* and *Omy 27*, respectively, were also consistent with genomic regions explaining more than 1% of the genetic variance for *P. salmonis* resistance [23].

Host defense against infection by *P. salmonis* seems to depend on many biological features and processes. For instance, this bacterium enters mainly through the skin and gills, thus, physical barriers can be considered an important defense mechanism against the pathogen. In addition, phagocytosis is a key process in the life cycle of *P. salmonis* and it is considered the primary mode of pathogenesis. The bacterium is internalized by clathrin-dependent endocytosis in phagocytic cells, and once it enters the cell, the cytoskeleton is significantly reorganized by altering myosins, actins, among others [15], In addition, several other processes are involved in the interaction between *P. salmonis* and the host, including kinase and helicase activity, lipid metabolism, inflammation, GTP hydrolysis, and the innate immune response [56]. Several candidate genes were found flanking significant SNPs for TD Several candidate genes were found flanking significant SNPs for TD. In Omy03, we found the cation-dependent mannose 6-phosphate receptor (CD-MPR), which works as a pattern recognition receptor that interacts with invading bacteria and triggers the expression of antimicrobial peptides against pathogens [67]. This gene was described in rainbow trout [68] with participation in lysosome formation [69,70]. Ref. [71] described that a decrease in the lysosomal response may be related to a host immune evasion strategy by *P. salmonis* in Atlantic salmon (*Salmo salar*). In the same chromosome, we found the pleckstrin homology domain-containing, family G (with RhoGef domain) member 6 (*PLEKHG6*), also known as myosin interacting guanine nucleotide exchange factor (MyoGEF) that participates in the regulation of apoptosis with the appearance of blisters and cytokinesis [72,73], and possible suppressor of carcinogenic tumors [74]. Ref. [75] identified the MyoGEF gene as a mediator of the immune response, cell proliferation and response against the ectoparasite *Neoparamoeba perurans* in Atlantic salmon.

In *Omy13*, we found the specific erythropoietin receptor (*EPOR*), a fundamental signaling agent in erythropoiesis [76,77]. The EPO:EPOR junction activates the JAK/STAT signaling pathway, which plays an important role in modulating oxidative stress, innate immunity and inflammation [78,79,80]. This gene may also act in non-hematopoietic tissues under normal or infection conditions [78], such as wound healing and cardiovascular protection [76]. Due to skin lesions, such as petechial hemorrhages and the presence of ulcers, caused by *P. salmonis* in salmonids, the EPOR could be a candidate gene involved in defense mechanisms. In a study developed by [81], the authors identified that heme metabolism was affected by the infection of *P. salmonis* in Atlantic salmon by a lack of sufficient amounts of iron in the head kidney, resulting in a decrease in the expression of erythrocyte-specific EPOR genes.

In Omy26, we found the N-acetylglucosamine-6-sulfatase-like (GNSL) gene, which is related to phagocytosis [82]; the CBFA2T3 gene, also known as ETO2, which is a crucial regulator for oncogenesis [83,84], inflammation [85,86] and also a hematopoietic co-repressor [87]; and high-mobility group box family member 3 (TOX3), which is a regulator of innate lymphoid cells, in particular, CD8+ cytotoxic T lymphocytes, which in turn has been shown to be activated by the ectoparasite *N.perurans* in Atlantic salmon.

In Omy27, we found the cytoskeletal protein smoothelin-like protein 2 (SMTNL2) expressed in skeletal muscle, which is activated by c-Jun N-terminal kinase (JNK), in response to cellular environmental stress, including inflammatory stimuli and apoptosis [88,89]. Ref. [23] found the Smtnl2 gene associated with *P. salmonis* resistance in the same rainbow trout population. The leucine-rich repeat-containing protein 75A-like gene (LRRC75A), located on this same chromosome, belongs to the long non-coding RNAs (lncRNAs) that play multiple vital roles during inflammatory processes caused by pathogens [90,91]; possibly acting as a critical mediator of the inflammasome, an essential pro-inflammatory precursor [92]. Unconventional myosin-Ic (Myo1C), known to play an important role in bacterial diseases by encoding a member of the strange myosin family of proteins, was also found in *Omy27* [93]. This gene is essential in lipid raft membrane recycling and proteins that regulate plasma membrane plasticity and pathogen entry [94]. An increase in Myo1c expression was identified upon infection with *Aeromonas hydrophila*, a Gram-negative bacterium, in Rohu carp (*Labeo rohita*) liver tissue [95]. The large neutral amino acid transporter small subunit 4 (LAT4) gene, also identified on chromosome 27, is essential in amino acid metabolism [96,97] and is involved in immune function [98,99]. In this same chromosome, we found the phosphatidylinositol transfer protein alpha (PITPNA) gene, essential for cell growth, organization of the cytoskeleton, metabolism, and apoptosis [100,101]. This gene was also previously identified to be associated with resistance to *P. salmonis* in rainbow trout [23]. The inositol polyphosphate 5-phosphatase K (INPP5K) gene also found in *Omy27*, participates in response to endoplasmic reticulum stress, organization of the cytoskeleton, renal osmoregulation, cell adhesion, and migration [102]. Loss of INPP5K may cause autophagy inhibition and lysosome depletion [103]. The INPP5K gene was identified to be involved in response to Aeromonas hydrophila infection in the European eel (*Anguilla anguilla*) [104]. F-box-only protein 39-like (FBXO39) participates in immune responses through the ubiquitin-proteasome system, promotes apoptosis of malignant cells [105], and has been described as an essential gene for animal behavior caused by stress in competition or fighting [106]. The X-linked inhibitor of apoptosis (XIAP)-associated factor 1 (XAF1), also found in *Omy27*, is a regulator of cytokinesis that considerably increases stress-induced apoptosis and decreases the invasive capacity of tumor cells [107,108]. XAF1 was identified as a gene involved in the primary responses to viruses with a pro-apoptotic function in Atlantic salmon [109,110], which could be considered a candidate gene for early response in microbial invasion in the host. In chromosome 27, we also found the bleomycin hydrolase gene (BLMH) that has a role in the regulation of inflammatory chemokines to initiate the immune response, wound healing [111], and skin integrity [112]. The expression of BLMH was found in the mucus of the skin of Greater amberjack (*Seriola dumerili*) infected with the ectoparasite *Neobenedenia girallae* [113]. In rainbow trout, BLMH expression has been identified as an innate response in the gill epithelial cell line to infection with ultraviolet-inactivated viral hemorrhagic septicemia virus [114]. The mediator of RNA polymerase II transcription subunit 19th gene (MED19), is an essential gene for the maintenance of white adipose tissue and adipogenesis. MED19 interacts with proliferator receptors activated peroxisome (PPARs) important in inflammatory processes [115]. The activation of PPARs decreases the expression of genes involved in oxidative stress, inflammation, migration, and cell growth [116]. Molecules that are not obviously related with processes which may be involved in host response against *P. salmonis*, were also identified in *Omy27*: Tektin 1 (TEKT1) has been shown to be related to sperm quality, cytoskeleton organization, and motility [117,118]; sodium-dependent serotonin transporter-like (SLC6A4) which is a gene that modulates many brain functions [119].

For BS, three essential candidate genes were found flanking the significant SNPs. In *Omy03*, we found the domain family protein 1 (TSC22D1) gene that participates as an early immune response molecule and that modulates the TGF-beta dependent signaling pathway [120], a tumor suppressor in cells [121,122], and induces cellular apoptosis and senescence [123]. These genes are from the TSC22 family, regulatory proteins with anti-inflammatory and immunosuppressive effects of interleukins and glucocorticoids [124]. In a transcriptome study in Atlantic salmon infected with *P. salmonis*, a decrease in the expression of TSC22D, from the TSC22 family, was shown in the head kidney [125]. In *Omy17*, we found the striatin-interacting protein one homolog (STRIP1) gene, which plays an important role in the migration and metastasis of cancer cells [126,127] and regulates the organization of the actin cytoskeleton [128]. In *Omy26*, we found a single gene called AP-2 complex subunit alpha-2 (AP2A2), involved in the activation of the leukocyte response, granulocytes, and myeloid cells involved in the immune response [129] and provides the first line of defense against a pathogen or virus [130].

Genes associated with both TD and BS were found in *Omy26*. Myotubularin-related protein 10 (MTMR10), which is associated with the protection of dendrites induced by oxidative stress or infection by pathogens and is present in several tissues was identified [131]. Nile tilapia exposed to ammonia showed a response in lipid metabolism-related to the MRM10 gene [132]. In Omy27, we found the protein Rap1 regulator of platelet activity [133,134]. It has been shown that (Rap1GAP2) is the only Rap1 GTPase activator protein in platelet segregation that is important in the immune system [135]. The expression of the immune-related gene Rap1GAP2 has been modulated in channel catfish (*Ictalurus punctatus*) after the infection with *Flavobacterium columnare* [136]. Ref. [137] observed high expression of this gene in both the spleen and kidney of giant grouper (*Epinephelus lanceolatus*) infected with spotted knifejaw iridovirus (SKIV). Slingshot phosphatases (SSH) are critical in controlling actin dynamics [138], especially the gene found on chromosome 27 called Slingshot phosphatase homolog 2 (SSH2), associated with inflammation. SSH2 is involved in the effective migration and invasion of tumor cells [139,140,141]. This gene could cause damage and autoinflammatory diseases due to excess neutrophils in healthy tissues associated with neutrophil chemotaxis [141].

We found 24 candidate genes that may be involved in *P. salmonis* resistance in rainbow trout, of which 14 have been identified in *Omy27.* Likewise, in [23], which used the same population, two genes were identified in common (PITPNA and SMTNL2) in *Omy27* for TD. [56] used a genome-wide association study analysis applying a Bayesian approach to investigate the genetic architecture of resistance to *P. salmonis* in farmed populations of salmonids, including rainbow trout. The authors identified nine candidate genes in Omy27 (SMTNL2, LRRC75A, unconventional myosin-Ic, LAT4, PITPNA, INPP5K, TEKT1, BLMH, and SLC6A4). These same genes were identified in our study for the TD phenotype. These candidate genes may be considered strong candidates to explain *P. salmonis* resistance variation in rainbow trout and are potentially relevant to test further genetics- or pharmacological-based strategies for controlling this disease in rainbow trout.

## 5. Conclusions

Resistance to the infectious disease caused by *P. salmonis* (SRS) has a complex polygenic inheritance architecture in rainbow trout. Here, we use imputed whole genome sequence data to refine the GWAS results for *P. salmonis* resistance in rainbow trout. We found 24 putative functional genes related to phagocytosis, innate immune response, inflammation, oxidative response, G-protein signaling pathway, lipid metabolism, and apoptotic processes, which may be related to variation in the host response against the bacterium, which is mainly found in a significant QTL found on chromosome 27. Further studies are needed to validate the results obtained in this study, especially related to candidate genes located on Omy27.

## Figures and Tables

**Figure 1 genes-14-00114-f001:**
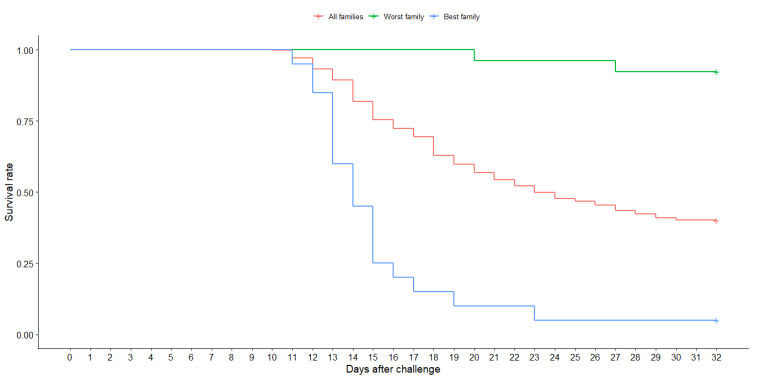
Kaplan–Meier survival curves of the average of the 102 full-sib families, the best and the worst family after an experimental challenge with *P. salmonis* in a breeding rainbow trout (*O. mykiss*) population.

**Figure 2 genes-14-00114-f002:**
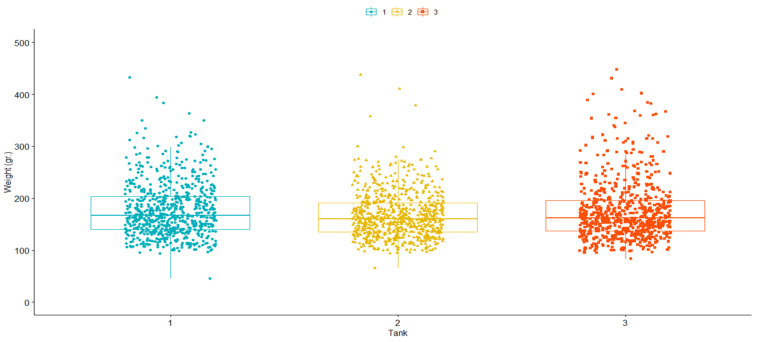
Boxplot for the covariate final weight (FW), post-challenge against *P. salmonis* in a rainbow trout (*O. mykiss*) breeding population.

**Figure 3 genes-14-00114-f003:**
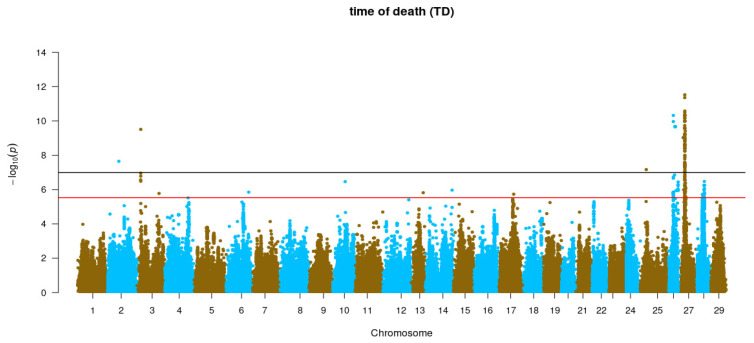
Genome-wide association analysis for resistance to *P. salmonis* in rainbow trout (*O. mykiss*) using WGS-imputed data. The black and red lines represent the genome-wide and chromosome-wide significance thresholds, respectively.

**Table 1 genes-14-00114-t001:** Quality control results of genotyping of 57K SNP panel, whole genome sequence (WGS) and Imputed WGS data for the rainbow trout population.

	$57K{\;}^{a}$	$WGS{\;}^{b}$	$Imputed\;WGS{\;}^{c}$
**Number of samples**	2130	102	2130
**Initial SNPs**	46,482	22,649,022	579,960
**Call-rate**	38,744	3,787,940	579,960
**Minor allele frequency**	37,672	2,599,032	537,784
**Hardy–Weinberg equilibrium**	31,215	2,382,000	488,979

^a^ Call-rate (0.90); MAF (0.01) y HWE ($10^{-8}$). ^b^ Call-rate (0.80); MAF (0.01) y HWE ($10^{-8}$). ^c^ Call-rate (0.80); MAF (0.05) y HWE ($10^{-12}$).

**Table 2 genes-14-00114-t002:** Estimates of variance components and heritability for *P. salmonis* resistance, defined as time to death (TD) and binary survival (BS), in a rainbow trout breeding population using whole-genome sequence imputed genotypes.

Trait/Parameter	$\sigma_{a}^{2}{\;}^{c}$	$\sigma_{e}^{2}{\;}^{d}$	$h^{2}\;(\mathrm{SE}){\;}^{e}$
TD ^a^	13.59	36.18	0.27 (0.02)
BS ^b^	0.045	0.134	0.25 (0.02)

^a^ Time to death. ^b^ Binary survival. ^c^ $\sigma_{a}^{2}$: additive genetic variance. ^d^ $\sigma_{e}^{2}$: residual variance. ^e^ $h^{2}$(SE): heritability (standard error).

**Table 3 genes-14-00114-t003:** Results of the gene-based analysis using MAGMA: Candidate genes related to resistance to *P. salmonis* in rainbow trout for time of death (TD) and binary survival (BS) traits.

CHR ^a^	START ^b^	STOP ^c^	*p* ^d^	Name ^e^	Trait ^f^	Function ^g^
3	15,087,418	15,094,391	$2.69\times 10^{-7}$	Cation-dependent mannose-6-phosphate receptor	TD	Lysosomal
3	15,408,170	15,722,916	$1.02\times 10^{-8}$	Pleckstrin homology domain containing, family G (with RhoGef domain) member 6	TD	Apoptosis
3	66,685,266	66,751,795	$2.11\times 10^{-6}$	TSC22 domain family protein 1	BS	Inflammation/Apoptosis
13	40,702,364	40,707,470	$1.51\times 10^{-6}$	Erythropoietin receptor	TD	Inflammation
17	48,001,343	48,015,360	$1.99\times 10^{-6}$	Striatin-interacting protein 1 homolog	BS	Organization of the cytoskeleton
26	12,843,404	12,851,300	$4.50\times 10^{-7}$	N-acetylgalactosamine-6-sulfatase-like	TD	Phagocytosis
26	12,858,495	12,910,113	$1.38\;\times 10^{-6}$	protein CBFA2T3	TD	Immune response
26	26,219,318	26,284,525	$5.78\times 10^{-6}$	TOX high mobility group box family member 3	TD	Immune response
26	18,051,513	18,080,240	$3.75\times 10^{-7}$	Myotubularin related protein 10	TD/BS	Lipid metabolism
26	21,925,171	21,942,499	$3.44\times 10^{-7}$	AP-2 complex subunit alpha-2	BS	Immune response
27	9,997,966	10,017,806	$5.39\times 10^{-7}$	Smoothelin protein 2	TD	Immune response
27	10,055,887	10,086,375	$1.28\times 10^{-7}$	Leucine-rich repeat-containing protein 75A	TD	Immune response
27	10,112,119	10,185,117	$6.46\times 10^{-10}$	Unconventional myosin-Ic	TD	Immune response
27	10,192,105	10,217,956	$2.07\times 10^{-10}$	Large neutral amino acids transporter small subunit 4	TD	Organization of the cytoskeleton
27	10,221,995	10,236,446	$3.08\times 10^{-10}$	Phosphatidylinositol transfer protein alpha	TD	Organization of the cytoskeleton
27	10,238,047	10,250,315	$5.91\times 10^{-7}$	Inositol polyphosphate 5-phosphatase K	TD	Organization of the cytoskeleton
27	10,252,627	10,257,723	$1.53\times 10^{-7}$	Tektin-1	TD	Organization of the cytoskeleton
27	10,257,875	10,260,754	$2.61\times 10^{-8}$	F-box only protein 39-like	TD	immune response
27	10,260,862	10,263,488	$2.62\times 10^{-8}$	XIAP-associated factor 1	TD	Apoptosis
27	10,384,815	10,405,300	$3.37\times 10^{-9}$	Bleomycin Hydrolase	TD	Immune response
27	10,409,313	10,439,943	$1.36\times 10^{-6}$	Sodium-dependent serotonin transporter	TD	Posttraumatic stress
27	10,606,714	10,608,776	$2.08\times 10^{-6}$	Mediator of RNA polymerase II transcription subunit 19a	TD	Inflammation
27	10,300,075	10,379,486	$1.23\times 10^{-6}$	RAP1 GTPase activating protein 2a	TD/BS	immune response
27	10,493,554	10,513,502	$1.55\times 10^{-6}$	Protein phosphatase Slingshot homolog 2	TD/BS	Inflammation

^a^ Chromosomal location. ^b^ Chromosomal start location of gene. ^c^ Chromosomal stop location of gene. ^d^ Gene level *p*-values. ^e^ gene name. ^f^ time to death (TD) and binary survival (BS). ^g^ Principal biological function.

## Data Availability

The datasets generated and analyzed during the current study are available upon request.

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
