# Peer review of "GWAS on Imputed Whole-Genome Sequence Variants Reveal Genes Associated with Resistance to *Piscirickettsia salmonis* in Rainbow Trout (*Oncorhynchus mykiss*)"

_genes, 2022, doi:10.3390/genes14010114_

Round 1
Reviewer 1 Report
Reviewer comments for Sanchez-Roncancio et al, ‘GWAS on imputed whole-genome sequence variants reveal genes associated with bacterial disease resistance in rainbow trout (Oncorhynchus mykiss)’, submitted to Genes (Manuscript ID genes-2065621).
The authors describe a GWAS for SRS resistance as assessed in a controlled challenge and using genotype data from a 57K SNP chip imputed to WGS. The authors have published extensively in this important arena of genomic analysis of disease resistance traits in aquaculture species.
The manuscript is generally well written and the data seem suitable for publication in the Journal pending consideration of the general and specific comments below.
General comment: The novel aspect of this study is the use of imputed genotypes to improve detection of genomic regions affecting SRS resistance (lines 90-95). However, the Discussion is essentially devoid of this aspect and instead simply describes the candidate genes the GWAS identified. The reader, therefore, seems to be left without an answer to the general question: how much, if any, did imputation help? Recommend the authors revise the Discussion to address this novel aspect of the study.
Line comments:
Line 1: minor comment, but in the title I would suggest using ‘resistance to Piscirickettsia salmonis’ or ‘resistance to salmon rickettsial syndrome’ instead of the generic ‘bacterial disease resistance’
Lines 24: the “and Omy 25;” seems incorrect; I believe Omy 25 should be in the running list of 5 regions associated with time of death.
Line 50: I know this gets a little tangential to the aim of the paper, but as discussed in the FIG review (reference #10) is it possible that the binary survival trait measured in this study is reflective of differences in tolerance rather than resistance?
Lines 105-111: This doesn’t make sense, it basically says the fish were tagged at 7 grams, reared for 7 months, moved to the research station, acclimated for 20 days, and then challenged at the same BW (7 grams) as when they were tagged. Please correct.
Lines 126-127: ‘department of agriculture of the United States USDA seems awkward, recommend changing to ‘USDA Agricultural Research Service’. Also, it should be the ‘National Center for Cool and Cold Water Aquaculture’ (missing the ‘Cool’).
Line 131: this section seems to be missing details regarding imputation
Line 149: For TD, was a value of 32 assigned to survivors?
Line 168: change ‘identify’ to ‘identified’
Line 181: this sentence is awkward as written, please revise
Lines 185-187: this sentence is awkward as written, please revise
Lines 248-249: shouldn’t it be 5 regions for TD and one for BS? What proportion of the genetic variance did these QTLs explain?
Lines 306-310: This sentence seems incomplete, what is being compared to previous studies here?
Lines 318-322: this sentence seems awkward/redundant, please revise.
Line 346: Here and throughout, it seems awkward to repeat ‘in the same chromosome we found…’ Perhaps consider reorganizing/grouping the discussion around a broader biological function (e.g., those given in lines 268-270) rather than chromosome by chromosome.
Line 355: is JNK given in table 3?
Line 420: perhaps replace the first ‘found in’ with ‘associated with’
Line 427: I’ll defer to the editor, but here and throughout the discussion it seems awkward to start a sentence like “[125] found expression…”
Author Response
Line 1: minor comment, but in the title I would suggest using ‘resistance to Piscirickettsia salmonis’ or ‘resistance to salmon rickettsial syndrome’ instead of the generic ‘bacterial disease resistance’
Response:Title was replaced by: “GWAS on imputed whole-genome sequence variants reveal genes associated with resistance to Piscirickettsia salmonis in rainbow trout (Oncorhynchus mykiss)”
Lines 24: the “and Omy 25;” seems incorrect; I believe Omy 25 should be in the running list of 5 regions associated with time of death.
Response: Corrected (L23 – L24): “GWAS revealed genome-wide significant quantitative trait loci (QTL) in Omy02, Omy03, Omy25, Omy26 and Omy27 for time to death and in Omy26 for binary survival.”
Line 50: I know this gets a little tangential to the aim of the paper, but as discussed in the FIG review (reference #10) is it possible that the binary survival trait measured in this study is reflective of differences in tolerance rather than resistance?
Response: Several studies consider the binary survival trait as a measure of resistance to infectious diseases in fish. It is difficult to discern whether it is resistance and/or tolerance and the presence of one does not exclude the other. However, tolerance is more related to the ability of the host to limit the impact of a given pathogen on the host performance. The best way to measure tolerance is to study the reaction norm slope of host performance regressed against individual's pathogen burden. Thus, binary survival may be more related to resistance than tolerance.
Lines 105-111: This doesn’t make sense, it basically says the fish were tagged at 7 grams, reared for 7 months, moved to the research station, acclimated for 20 days, and then challenged at the same BW (7 grams) as when they were tagged. Please correct.
Response:We dropped this wrong value for mean weight from this section (L109).
Lines 126-127: ‘department of agriculture of the United States USDA seems awkward, recommend changing to ‘USDA Agricultural Research Service’. Also, it should be the ‘National Center for Cool and Cold Water Aquaculture’ (missing the ‘Cool’).
Response: Corrected (L123-L126): “Individual genotypes were obtained using a commercial 57K SNP chip (Affymetrix Axiom) developed by the National Center for Cool and Cold Water Aquaculture of the USDA Agricultural Research Service [54]”.
Line 131: this section seems to be missing details regarding imputation
Response: We added this section about genotype imputation (L147-L158): ” 2.4 Imputation to whole genome sequence level
For the validation of the imputation, five cross-validation sets were formed using only the 102 re-sequenced individuals, divided into 20% validation (approximately 20 animals per group) and 80% reference (approximately 80 animals per group) sets. The validation animals had only the 57K SNPs in common with the genotyped animals (challenged animals) and the remaining SNPs were masked to perform genotype imputation. Reference animals had information at WGS level including only SNPs that passed QC. FImpute v3.0 [57]was used to perform all genotype imputations. Accuracy was assessed using the r2 (Pearson's squared correlation) statistics between observed and imputed genotypes, and SNPs with low imputation accuracy (lower than 0.8) were removed from the final set of imputable SNPs. Finally, the challenged animals, genotyped with 57k SNPs, were imputed using only validated SNPs with accuracy above 0.8.”
Line 149: For TD, was a value of 32 assigned to survivors?
Response:Yes, individuals that survived through all challenge had 32 at time of death.
Line 168: change ‘identify’ to ‘identified’
Response: line 179 ,Modified.
Line 181: this sentence is awkward as written, please revise
Response: Corrected (L192-L193): “Similar mean values were observed for TD: 24.55 (SD = 7.60); 23.5 (SD = 7.82) and 22.73 (SD = 8.08) days for tanks 1, 2, and 3, respectively.”
Lines 185-187: this sentence is awkward as written, please revise
Response: Corrected (L196-L198): “For final weight (FW), mean values were 177.5 g (SD = 52.05); 167.3 g (SD = 44.90) and 176.2 g (SD = 58.62) for tanks 1, 2 and 3, respectively (Figure 2).”
Lines 248-249: shouldn’t it be 5 regions for TD and one for BS? What proportion of the genetic variance did these QTLs explain?
Response: Yes, TD and BS were exchanged, we corrected it (L258-L259).
We added this information about the phenotypic variance explained by these QTLs (L264-L267): “All QTLs found here explained a low phenotypic variance for resistance against P. salmonis, reinforcing the fact that this trait is under polygenic control in rainbow trout (Supplementary 1, Table S1 and Table S2).”
Lines 306-310: This sentence seems incomplete, what is being compared to previous studies here?
Response: We corrected this sentence (L319-L323): “Although imputation or genotyping errors may generate false positives of association, the stringent filters used for the QC of true and WGS-imputed genotypes aimed at decreasing the probability of false positive detection. In addition, imputation results are in agreement with previous studies which performed genotype imputation to WGS in rainbow trout and other aquaculture species [40,41,42].”
Lines 318-322: this sentence seems awkward/redundant, please revise.
Response: We corrected this sentence (L335-L338): “Several candidate genes were found flanking significant SNPs for TD. In Omy03, we found the cation-dependent mannose 6-phosphate receptor (CD-MPR), which works as a pattern recognition receptor that interacts with invading bacteria and triggers the expression of antimicrobial peptides against pathogens [63].”
Line 346: Here and throughout, it seems awkward to repeat ‘in the same chromosome we found…’ Perhaps consider reorganizing/grouping the discussion around a broader biological function (e.g., those given in lines 268-270) rather than chromosome by chromosome.
Response: We corrected this sentence (L361-L366): “In Omy26, we found the N-acetylglucosamine-6-sulfatase-like (GNSL) gene, which is related to phagocytosis [78]; the CBFA2T3 gene, also known as ETO2, which is a crucial regulator for oncogenesis [79,80], inflammation [81,82] and also a hematopoietic co-repressor [83]; and high-mobility group box family member 3 (TOX3), which is a regulator of innate lymphoid cells, in particular, CD8+ cytotoxic T lymphocytes, which in turn has been shown to be activated by the ectoparasite Neoparamoeba perurans in Atlantic salmon.”
Line 355: is JNK given in table 3?
Response: We corrected this sentence (L368-L371): “In Omy27, we found the cytoskeletal protein smoothelin-like protein 2 (SMTNL2) expressed in skeletal muscle, which is activated by c-Jun N-terminal kinase (JNK), in response to cellular environmental stress, including inflammatory stimuli and apoptosis [84,85]”
Line 420: perhaps replace the first ‘found in’ with ‘associated with’
Response: We corrected this sentence (L433-L436): “Genes associated with in both TD and BS were found in Omy26. Myotubularin related protein 10 (MTMR10), which is associated with the protection of dendrites induced by oxidative stress or infection by pathogens and is present in several tissues was identified [127].”
Line 427: I’ll defer to the editor, but here and throughout the discussion it seems awkward to start a sentence like “[125] found expression…”
Response: We corrected this sentence (L436-L441): “Genes associated with both TD and BS were found in Omy26. For instance, we identified myotubularin related protein 10 (MTMR10), which is associated with the protection of dendrites from the damage induced by oxidative stress and infection [127]. Nile tilapia exposed to ammonia shown a response in lipid metabolism-related to the MRM10 gene [128]. In Omy27, we found the protein Rap1 regulator of platelet activity [129,130]. It has been shown that (Rap1GAP2) is the only Rap1 GTPase activator protein in platelet segregation that is important in the immune system [131]. The expression of the immune-related gene Rap1GAP2 has been modulated in channel catfish (Ictalurus punctatus) after the infection with Flavobacterium columnare [132].
Reviewer 2 Report
General comments
Pisirickettsiosis is indeed the main bacterial infectious disease in the salmon industry and is only relatively under control due to the absence of effective vaccines. There are several studies regarding genetic resistance in salmonids species, but the problem persists. Any new knowledge generated to optimize the control of SRS will always be welcome, but the present manuscript requires some improvements.
Specific comments
L1: Why not be direct in the title and to indicate “piscirickettsiosis” instead “bacterial disease”?
L16-L17 I suggest rephrasing like this …. “Here, we conducted GWAS to assess resistance against the facultative intracellular bacterium Piscirickettsia salmonis, which affects farmed rainbow trout, Oncorhynchus mykiss, in Chile.”
L19: what challenge model did you use? I suggest you should be to indicate it here.
L19: Use “P. salmonis” instead “bacteria”
L24-L25: I suggest changing "24 genes" to "Twenty-four (24) candidate genes...".
L35: I suggest using the formal name "piscirickettsiosis" and rephrasing something like this: "...piscirickettsiosis or salmon rickettsial syndrome (SRS) caused by Piscirickettsia salmonis.” The rest of the sentence should be reedited.
L37-L38: This statement contains a major error because the official data indicates that for rainbow trout specie, 18.8% of total mortality in 2021 was attributable to infectious diseases, but only 29.7% of this was specifically attributable to SRS. Thus, the SRS-attributed mortality in 2021 was only 5.6% of the TOTAL mortality registered for the specie.
L40 and L41: References missed.
L42: There are updated statistics now. I suggest using them.
L43-L46: Many references missed.
L48: Several references are missing from this statement.
Rozas-Serri 2017
Rozas-Serri 2018a
Rozas-Serri 2018b
Rozas-Serri 2019
Rozas-Serri 2022
Figueroa 2021
Figueroa 2022
Etc
L49-L50: How could you summarize the results of these studies in practice? What has been the concrete contribution to SRS control in Chile?
L52-L60 I suggest being more direct in this description.
L61-L78 I suggest being more direct in this description. This section is to introduce your work and to give support to your hypothesis. It is not a review. The introduction should be more direct to be more understandable and easier to follow.
L90: I suggest that a more concrete message and a better explanation of what is the difference of this work with respect to what has already been published and, most importantly, what does it mean in practice that this study is carried out using a different technical approach? what is expected from the application of this new information?
L109: Why didn't you include PRV in this health exam? health screening?
L111: Why did you use the IP challenge model? The coexistence challenge model should be more advisable considering the normal pathway of disease transmission via horizontal transmission.
L112: Use “P. salmonis” instead of “P. Salmonis” and make sure it is maintained throughout the manuscript.
L116: How did you euthanize the fish?
L178: In general, the results should not be repeated in the text and in the tables.
L286: Authors should discuss starting immune modulation directed by P. salmonis in fish to better understand their genetic resistance results. In this context, several references are missed. In general, the biological interpretation and possible practical applications are weak. I suggest that the discussion be improved.
Author Response
L1: Why not be direct in the title and to indicate “piscirickettsiosis” instead “bacterial disease”?
Response: Title was replaced by: “GWAS on imputed whole-genome sequence variants reveal genes associated with resistance to Piscirickettsia salmonis in rainbow trout (Oncorhynchus mykiss)”
L16-L17 I suggest rephrasing like this …. “Here, we conducted GWAS to assess resistance against the facultative intracellular bacterium Piscirickettsia salmonis, which affects farmed rainbow trout, Oncorhynchus mykiss, in Chile.”
Response: We corrected this sentence (L16-L18): “Here, we conducted GWAS to assess resistance against the facultative intracellular bacterium Piscirickettsia salmonis, which affects farmed rainbow trout, Oncorhynchus mykiss, in Chile. We used imputed genotypes at the sequence level and searched for candidate genes located in genomic regions associated with the trait.
L19: what challenge model did you use? I suggest you should be to indicate it here.
Response: We used a challenge by intraperitoneal injection (see below).
L19: Use “P. salmonis” instead “bacteria”
Response: We corrected this sentence (L19-L21): A total of 2130 rainbow trout were intraperitoneally challenged with the P. salmonis under controlled conditions and genotyped using a 57K single nucleotide polymorphism (SNP) panel.
L24-L25: I suggest changing "24 genes" to "Twenty-four (24) candidate genes..."
Response: We corrected this sentence (L25): Twenty-four (24) candidate genes
L35: I suggest using the formal name "piscirickettsiosis" and rephrasing something like this: "...piscirickettsiosis or salmon rickettsial syndrome (SRS) caused by Piscirickettsia salmonis.” The rest of the sentence should be reedited.
Response: We corrected this sentence (L35-L37): “The production of this species is highly affected by infectious diseases, such as piscirickettsiosis or salmon rickettsiosis syndrome (SRS) caused by Piscirickettsia salmonis, producing the greatest economic losses".
L37-L38: This statement contains a major error because the official data indicates that for rainbow trout specie, 18.8% of total mortality in 2021 was attributable to infectious diseases, but only 29.7% of this was specifically attributable to SRS. Thus, the SRS-attributed mortality in 2021 was only 5.6% of the TOTAL mortality registered for the specie.
Response: We updated this sentence with new statistics (L37-L39) ”For example, for the first semester of 2022, 12% of total mortality was attributed to infectious diseases, and within this category 70.4% was specifically attributed to SRS [3]
L40 and L41: References missed. L42: There are updated statistics now. I suggest using them.
L43-L46: Many references missed. L48: Several references are missing from this statement.
Rozas-Serri 2017Rozas-Serri 2018aRozas-Serri 2018bRozas-Serri 2019Rozas-Serri 2022Figueroa 2021Figueroa 2022
Response: We included these references as suggested by the reviewer (L39-L51):
Prophylactic and control strategies against P. salmonis infection have been based mainly on vaccines and antibiotic administration during disease outbreaks. However, both approaches show low effectiveness in controlling SRS in field conditions [4,5]. During the first semester of 2022, 94.8% of the antibiotics administered in salmonids in Chile were for the treatment of P. salmonis infection [6].The use of antibiotics might result in resistant bacterial strains, not only reducing the efficacy of treatments and affecting the success and sustainability of aquaculture but also represents an important threat to public health [7,8].The limited efficacy of vaccines on reducing mortalities in sea-cages might be due to the variation in the host immune response between families and the complexity of the P. salmonis-host interaction [9,10,11] .An alternative strategy to reduce the detrimental effects of this disease in the long-term, is the genetic selection of fish more resistant to P. salmonis infection [13,14,15].
L49-L50: How could you summarize the results of these studies in practice? What has been the concrete contribution to SRS control in Chile?
Response: We added the following sentence to address this point:
L 51-53: “In fact, the most important salmon and trout breeding companies in Chile currently offer SRS resistant eggs within their products, which are generated from families which have shown superior survival in either experimental or field conditions.”
L52-L60 I suggest being more direct in this description.
L61-L78 I suggest being more direct in this description. This section is to introduce your work and to give support to your hypothesis. It is not a review. The introduction should be more direct to be more understandable and easier to follow.
Response: This paragraph is focused on the description of two approaches that can be performed using genomic information, GWAS and Genomic selection, describing the latter in these lines. We did slight modification to be clearer:
L62-71: “With the advances in genomics of aquatic organisms in recent years [15], it is now possible to include information from dense molecular markers (e.g., single nucleotide polymorphisms (SNPs), in aquaculture breeding programs [16]. Genomic information has two main applications for breeding programs: genomic selection and genome-wide association studies (GWAS). Genomic selection allows the estimation of genomic breeding values of selection candidates with greater accuracy than pedigree-based methods [17], which accelerate the genetic progress for the trait of interest. For example, the effect of genomic selection in increasing the accuracy on the estimation of the genetic merit for resistance to P. salmonis infection has been demonstrated in Atlantic Salmon [18], coho Salmon [19] and rainbow trout [15, 20].”
L90: I suggest that a more concrete message and a better explanation of what is the difference of this work with respect to what has already been published and, most importantly, what does it mean in practice that this study is carried out using a different technical approach? what is expected from the application of this new information?
Response: We modified the following paragraph to clarify the message that here we aimed at increasing resolution and statistical power by using (ten times) higher-density (x10) of genotypes than previous studies by using imputed whole-genome sequences:
L90-99: Potential QTLs associated with P. salmonis resistance have been found in rainbow trout and other salmonid species [18,19,20,52,53]. For instance, resistance to P. Salmonis in-fection was determined to be a polygenic trait in rainbow trout, Atlantic salmon and coho salmon, with few QTL explaining a moderate proportion of the genetic variance for the trait [19,20,52]. All these studies were performed using medium-density SNP panels, including tens of thousands of markers, making it difficult to map genomic re-gions with high accuracy. This study aimed to perform GWAS for resistance against SRS using genotype data imputed to high-density SNP information (i.e. WGS) for increasing statistical power and resolution in the detection of genomic regions and putative genes controlling the trait.
L109: Why didn't you include PRV in this health exam? health screening?
Response: Because, there was not evidence of previous presence of PRV in the studied breeding population.
L111: Why did you use the IP challenge model? The coexistence challenge model should be more advisable considering the normal pathway of disease transmission via horizontal transmission.
Response: We agree that cohabitation model could be more similar to the natural way of transmission of the disease. However, IP challenges have been extensively used to assess genetic variation for resistance to SRS given the high level of replication between experiments. In addition, genetic correlation between IP and cohabitation challenges have been shown to be > 0.8 (Martinez et a.., 2014; Dettleff et al., 2015), indicating that resistance to P. salmonis experimentally evaluated by IP or cohabitation are similar traits, from the genetic perspective.
V. Martinez, Genomic selection applied to Piscirickettsia salmonis resistance in Chilean Atlantic salmon, in: International Plant & Animal Genome XXII, January 11e15, 2014. San Diego, USA.
Dettleff, P., Bravo, C., Patel, A., & Martinez, V. (2015). Patterns of Piscirickettsia salmonis load in susceptible and resistant families of Salmo salar. Fish & Shellfish Immunology, 45(1), 67-71.
L112: Use “P. salmonis” instead of “P. Salmonis” and make sure it is maintained throughout the manuscript.
Response: All corrected
L116: How did you euthanize the fish?
Response: Fish were anesthetized with tricaine methanesulfonate (TMS) (0.1 g/L dosage) prior to any handling event. Fish were euthanized with a lethal dose of TMS.
L178: In general, the results should not be repeated in the text and in the tables.
Response: We corrected this paragraph (L224-L237)
“A 57K SNP chip was used for genotyping 2130 challenged rainbow trout. Initial quality control (QC) of genotypes was applied with Affymetrix Axiom software according to default settings. Additional QC filters were sequentially applied, discarding 17.6% SNPs by call-rate, 2.7% SNPs by MAF and 17.14% by HWE, leaving a total of 31,275 SNPs available for downstream analyses. For the genotypes from WGS data, 22,649,022 SNPs from the 102 samples were obtained after SNP calling. The call-rate filter eliminated about 83.3% of the initial SNPs. MAF and HWE filters discarded 5.2% and 1% of SNPs, respectively, with a remaining list of 2,382,000 SNPs available for downstream analysis (a total of 10.5% out of the initial number of SNPs). After imputation validation, 24.3% of the SNPs had an r² value equal or greater than 0.80 and were used as the reference data set for the definitive imputation round. A post-imputation quality control (HWE < p-value > 10^(-8) and MAF <0.05) resulted in a total of 488,979 available imputed SNPs for the 2,130 individuals reported as imputed WGS genotypes (Table 1)”
L286: Authors should discuss starting immune modulation directed by P. salmonis in fish to better understand their genetic resistance results. In this context, several references are missed. In general, the biological interpretation and possible practical applications are weak. I suggest that the discussion be improved.
Response: We added the following lines to improve discussion:
“Host defense against P. salmonis infection appears to depend on many biological processes, including kinase and helicase activity, lipid metabolism, cytoskeletal dynamics, inflammation, GTP hydrolysis, and innate immune response”
“Functional testing of the putative genes involved in P. salmonis resistance and their both individual and joint role in the trait will be facilitated by collaborative international research initiatives (e.g. Functional Annotation of All Salmonid Genomes).”
Reviewer 3 Report
1. The chromosomal significance threshold (0.05/number of SNPs/number of chromosomes) (Line 170), the description may be wrong, please modify.
2. GWAS was performed using a linear mixed animal model implemented for two trait definitions: time to death (TD) and binary sur-149 vival (BS). Whether the gwas analysis model selection is appropriate, please give evidence.
3. The estimated heritability for TD and BS using WGS-imputed genotypes were 0.27 ± 288 0.02 and 0.25 ± 0.02, respectively. These estimates were lower than those presented. However, there are significant loci in the results of GWAS analysis, which is somewhat puzzling, please give an explanation.
Author Response
1. The chromosomal significance threshold (0.05/number of SNPs/number of chromosomes) (Line 170), the description may be wrong, please modify.
Response: We corrected this sentence (L181) (0.05/number of SNPs on each chromosome)
2. GWAS was performed using a linear mixed animal model implemented for two trait definitions: time to death (TD) and binary sur-149 vival (BS). Whether the gwas analysis model selection is appropriate, please give evidence.
Response:
From a goodness-of-fit perspective, the most appropriate statistical approaches to adjust binary survival (BS) and time to death (TD) variables might be threshold and proportional-hazard frailty (PHF) models, respectively (Yáñez et al., 2014). Threshold models account for the dichotomic nature for BS while PHF models account for censored data for TD. However, from a predictive-ability perspective, these models are expected to work similar to ordinary linear mixed animal models for both traits (Yáñez et al., 2014). Regarding GWAS, BS has been previously fitted using threshold models using frequentist and Bayesian approaches in former studies from our group (Correa et al., 2015; Barría et al., 2018). Simultaneous fitting of regular linear mixed animal models for TD in the same previous studies have shown consistency of results between traits (BS and TD), suggesting that data censoring is not considerably affecting statistical power in GWAS for TD. Based on these results, here, we used a regular linear mixed animal model to perform GWAS for TD, and also the same approach to fit BS. The results between methods were somewhat consistent, however, we lost statistical power when we did not account for the binary nature of BS (Figure 3). Thus, BS might be better assessed by using a threshold model; nevertheless, this approach is not yet implemented in the software used in this study (GCTA).
Yáñez, J. M., Bangera, R., Lhorente, J. P., Oyarzún, M., & Neira, R. (2013). Quantitative genetic variation of resistance against Piscirickettsia salmonis in Atlantic salmon (Salmo salar). Aquaculture, 414, 155-159.
Correa, K., Lhorente, J. P., López, M. E., Bassini, L., Naswa, S., Deeb, N., ... & Yáñez, J. M. (2015). Genome-wide association analysis reveals loci associated with resistance against Piscirickettsia salmonis in two Atlantic salmon (Salmo salar L.) chromosomes. BMC genomics, 16(1), 1-9.
Barría, A., Christensen, K. A., Yoshida, G. M., Correa, K., Jedlicki, A., Lhorente, J. P., ... & Yáñez, J. M. (2018). Genomic predictions and genome-wide association study of resistance against Piscirickettsia salmonis in coho salmon (Oncorhynchus kisutch) using ddRAD sequencing. G3: Genes, genomes, genetics, 8(4), 1183-1194.
3. The estimated heritability for TD and BS using WGS-imputed genotypes were 0.27 ± 288 0.02 and 0.25 ± 0.02, respectively. These estimates were lower than those presented. However, there are significant loci in the results of GWAS analysis, which is somewhat puzzling, please give an explanation.
Response:
The GWAS show the strength of the statistical association between the SNPs and the phenotype. This is not directly related to heritability since a SNP may be associated but explain low genetic variance. Within the document we show the phenotypic variance explained by markers was low. In other words, all QTLs found here explained a low phenotypic variance for resistance against P. salmonis, reinforcing the fact that this trait is under polygenic control in rainbow trout (Supplementary 1, Table S1 and Table S2). The differences on the estimated heritability for TD and BS between this and previous studies might be due to the fact that here we used SNP-heritability estimation (i.e. the fraction of the phenotypic variance explained by additive effects of the genetic variants used here), which has not been used in other studies. In addition, we did not discard the possibility that imputation error may have had an impact in the estimation of genetic parameters. See below:
L302-312: “The estimated SNP-heritability for TD and BS using WGS-imputed genotypes were 0.27 ± 0.02 and 0.25 ± 0.02, respectively. These estimates were lower than those presented by [20], which were calculated in the same population using a different statistical approach implemented in BLUPF90 software, with heritability values of 0.48 ± 0.04 and 0.34 ± 0.04 for TD and BS, respectively. Here we used a different density of SNPs compared with the previous study, which by imputation error may have had an impact in the estimation of genetic parameters. However, the results obtained are similar to those obtained when pedigree and genomic information was simultaneously used to estimate the genetic variance for resistance against P. salmonis in rainbow trout [15,20]. In general, our results confirm that there is a significant additive-genetic com-ponent involved phenotypic variance for P. salmonis resistance.”
Round 2
Reviewer 2 Report
- It is Salmonid Rickettsial Syndrome (no "Rickettsiosis")
- In this sentence: "For example, for the first semester of 2022, 12% of total mortality was attributed to infectious diseases, but only 70.4% was specifically attributed to an SRS[3]" Why for example? Why ONLY? Do you think that 70% of 12% is little? I suggest you speak directly about the fact that 8.5% of the total mortality was attributed to SRS.
- It is P. salmonis, no P. Salmonis
- The P. salmons-fish biological and immunological interaction it poorly discussed
- Important references are missed
- If you wrote Infectious Salmon Anaemia virus (ISAV) then you should write "IPN virus" using the same description = Infectious Pancreatic Necrosis virus
- What happened to Piscine Orthoreovirus PRV?? It is very widespread in Chile but you did not verify it. I suggest including that you did not run PCR for PRV.
- I suggest including some discussion of IP versus cohabitation challenge models using P. salmonis
Author Response
It is Salmonid Rickettsial Syndrome (no "Rickettsiosis")
Corrected (L35-L36)
- In this sentence: "For example, for the first semester of 2022, 12% of total mortality was attributed to infectious diseases, but only 70.4% was specifically attributed to an SRS[3]" Why for example? Why ONLY? Do you think that 70% of 12% is little? I suggest you speak directly about the fact that 8.5% of the total mortality was attributed to SRS.
Corrected (L37): “the first semester of 2022, 8.5% of the total mortality was attributed to SRS”
- It is P. salmonis, no P. Salmonis:
Corrected
- The P. salmons-fish biological and immunological interaction it poorly discussed:
This fish-P. salmonis interaction is widely described in a recent review by Rozas-Serri (2022), which is referenced in the discussion. We extended a previous paragraph to include a brief description of the factors involved in this interaction.
Rozas-Serri, M. (2022). Why Does Piscirickettsia salmonis Break the Immunological Paradigm in Farmed Salmon? Biological Context to Understand the Relative Control of Piscirickettsiosis. Frontiers in Immunology, 13, 856896-856896.
Corrected (L339-L345): “Host defense against infection by P. salmonis seems to depend on many biological features and processes. For instance, this bacterium enters mainly through the skin and gills, thus, physical barriers can be considered as an important defense mechanisms against the pathogen. In addition, phagocytosis is a key process in the life cycle of P. salmonis and it is considered the primary mode of pathogenesis. The bacterium is internalized by clathrin-dependent endocytosis in phagocytic cells, and once it enters the cell, the cytoskeleton is significantly reorganized by altering myosins, actins, among others (Rozas 2022). In addition, several other processes are involved in the interaction between P. salmonis and the host, including kinase and helicase activity, lipid metabolism, inflammation, GTP hydrolysis, and the innate immune response [62].”
- Important references are missed
We included these references as suggested by the reviewer (L46-L57).
- If you wrote Infectious Salmon Anaemia virus (ISAV) then you should write "IPN virus" using the same description = Infectious Pancreatic Necrosis virus
Corrected (L107-L108): “ Random sampling was performed to check for Flavobacterium spp., Infectious Pancreatic Necrosis virus (IPNV) , infectious salmon anemia virus (ISAV), and Renibacterium salmoninarum using qRT-PCR “
-What happened to Piscine Orthoreovirus PRV?? It is very widespread in Chile but you did not verify it. I suggest including that you did not run PCR for PRV.
Corrected (L109-L110): “and PCR for Piscine Orthoreovirus (PRV) was not performed”
-I suggest including some discussion of IP versus cohabitation challenge models using P. salmonis
We added the following lines to improve discussion (L303-L307) “IP challenges have been extensively used to assess genetic variation for resistance to SRS given the high level of replication between experiments. In addition, genetic correlation between IP and cohabitation challenges have been shown to be > 0.8 (Martinez et a.., 2014; Dettleff et al., 2015), indicating that resistance to P. salmonis experimentally evaluated by either IP or cohabitation are similar traits, from the quantitative genetic perspective”